# A Fractal Curve-Inspired Framework for Enhanced Semantic Segmentation of Remote Sensing Images

**DOI:** 10.3390/s24227159

**Published:** 2024-11-07

**Authors:** Xinhua Wang, Botao Yuan, Zhuang Li, Heqi Wang

**Affiliations:** 1School of Computer Science, Northeast Electric Power University, Jilin 132012, China; 2202100980@neepu.edu.cn (B.Y.); liz@neepu.edu.cn (Z.L.); 2Changchun Institute of Optics, Fine Mechanics and Physics, Chinese Academy of Sciences, Changchun 130033, China; wanghq@ciomp.ac.cn

**Keywords:** remote sensing images, bilinear interpolation, fractal curve, gather layers, encoder–decoder, semantic segmentation

## Abstract

The classification and recognition of features play a vital role in production and daily life; however, the current semantic segmentation of remote sensing images is hampered by background interference and other factors, leading to issues such as fuzzy boundary segmentation. To address these challenges, we propose a novel module for encoding and reconstructing multi-dimensional feature layers. Our approach first utilizes a bilinear interpolation method to downsample the multi-dimensional feature layer in the coding stage of the U-shaped framework. Subsequently, we incorporate a fractal curve module into the encoder, which aggregates points on feature maps from different layers, effectively grouping points from diverse regions. Finally, we introduce an aggregation layer that combines the upsampling method from the UNet series, employing the multi-scale censoring of multi-dimensional feature map outputs from various layers to efficiently capture both spatial and feature information. The experimental results across diverse scenarios demonstrate that our model achieves excellent performance in aggregating point information from feature maps, significantly enhancing semantic segmentation tasks.

## 1. Introduction

In recent years, remote sensing images have found widespread applications in transportation analysis and management [1], urban area monitoring and planning [2,3], and disaster monitoring and avoidance [4]. This has sparked considerable interest in utilizing remote sensing images for feature classification [5,6] and distribution, with the delineation of areas and accurate detection of changes in feature coverage [7,8,9] being of paramount importance. These factors, along with limitations in spatial [10] and spectral resolution due to acquisition equipment, significantly impact the initial stages of research and the overall experimental process [11].

Traditionally, the feature classification of remote sensing images relied on shallow features extracted through manual labeling or machine learning. For instance, Andres et al. proposed a threshold-based collaborative segmentation method for buildings, employing multi-thresholding for single-target segmentation [12]. Lakshmi et al. advanced this approach by utilizing object edges as constraints, proposing an image segmentation method based on edge detection using differential operators [13].

Deep learning models have since emerged as powerful alternatives, effectively extracting deep feature information from images. These models learn image features through neural networks, establishing end-to-end frameworks for image-to-mask and pixel-to-pixel mapping relationships. This approach has garnered significant attention in remote sensing image segmentation.

Jonathan Long et al. implemented a semantic segmentation model using convolutional neural networks, replacing fully connected layers with convolutional layers to achieve image-to-mask mapping through encoding and decoding. However, this method faced issues of detail loss due to fixed receptive fields [14]. Vijay Badrinarayanan et al. proposed the SegNet model, utilizing pooled indexes to preserve image contour information while effectively reducing model parameters [15].

Kaiming He et al. introduced Mask R-CNN, employing ROIAlign [16] based on bilinear interpolation to address the inaccurate localization caused by the rounding operation in ROIPool [17]. Olaf Ronneberger et al. incorporated the multi-scale concept of skip connections into the FCN model [18], leading to the development of the UNet, UNet++ [19], and UNet3+ [20] series, all based on encoding–decoding U-shaped frameworks.

Concurrently, Liang-Chieh Chen et al. replaced standard convolution with dilated convolution, discarded fully connected layers, and modified the last two pooling layers in VGG16 (proposed by Karen Simonyan et al.) to mitigate downsampling issues [21]. This work led to the successive development of DeeplabV1 [22], DeeplabV2 [23], DeeplabV3 [24], and DeeplabV3+ [25] models, with DeeplabV2 notably contributing the flexible use of atrous convolution and atrous spatial pyramid pooling (ASPP) while abandoning CRF modules.

Qin X et al. employed a nesting approach, proposing the U^2^Net model that constructs a large U-shaped frame using smaller U-shaped frames as modules [26]. Valanarasu J M J et al. emphasized the importance of blocking local dependencies, introducing this concept in their UNeXt model [27]. Alrfou K et al. explored the fusion of a Transformer with UNet, resulting in the GCtx-Unet model that incorporates a Transformer architecture [28].

The semantic segmentation of remote sensing image multi-classification based on deep learning networks relies heavily on the effective representation of image feature information. However, interference from cluttered backgrounds, lighting conditions, and cloud cover makes it challenging for deep learning networks to capture the essence of edge features, resulting in the blurred edge segmentation of image objects. The introduction of fractal curves offers a solution by wandering through image pixels, effectively straightening and reconstructing the original image pixel arrangement.

Fractal curves, from their conceptual origins in the 17th century to the first concrete examples, have evolved into a series of classical forms [29], such as the Koch snowflake, Sierpinski series, and Mandelbrot set. These graphs exhibit unique self-similar structures with infinite details. At their core, they consist of discrete points capable of traversing space, finding applications in various fields. These include industrial products like fractal fluid devices [30], antennas [31], capacitors [32], and wave-absorbing materials [33], as well as computer sciences for data compression and image processing [34]. They also aid in data dimensionality reduction by leveraging the invariance of spatial mappings [35]. Fractal curves and their discrete approximations provide a mapping between one- and two-dimensional spaces that preserves localization, ensuring that similar points remain similar after transformation.

The models mentioned above are partly from medicine and partly from the field of remote sensing. Although medical semantic segmentation and remote sensing semantic segmentation differ in their application domains, they share a close correlation in their core methodologies. Both aim to identify and segment specific targets or regions from complex image data. In medical semantic segmentation, the objectives may include the recognition of tumors, blood vessels, or other significant biomarkers, whereas in remote sensing semantic segmentation, the goal might be to distinguish between different types of land cover, such as forests, urban areas, or water bodies. The core approach of both technologies relies on deep learning models to extract features from images. Additionally, when processing images, both need to consider the spatial resolution and dimensions of the images. In medical imaging, this means that models require extensive training on a vast array of image data to achieve classification of individual pixels within the images. Consequently, there is a potential for mutual inspiration and cross-fertilization in their technical implementation and algorithmic design.

Inspired by this theory, we propose FCUNet, a remote sensing image segmentation network that incorporates fractal curves to address over-segmentation and edge blurring between image objects, thereby improving intra-class cohesion and enhancing inter-class separability.

Our specific contributions are as follows:(1)We propose downsampling the image using bilinear interpolation to obtain centroid weights through trainable relative distances.(2)We introduce an aggregation layer to extract feature map channels at different scales and merge them with peer scales.(3)We utilize fractal curves to rearrange feature map points on the image, enhancing pixel aggregation.(4)We propose a new metric to evaluate the edge adhesion coefficient between objects after semantic segmentation.(5)We validate our approach on WHDLD, Potsdam, and Vaihingen remote sensing image datasets. The ablation experimental results demonstrate the effectiveness of the fractal curve module, with our proposed FCUNet model outperforming four original benchmark models.

## 2. Materials and Methods

Remote sensing image datasets face several challenges during processing, such as cluttered backgrounds, lighting variations, and cloud effects, which hinder traditional methods from achieving fine-grained feature extraction. This, in turn, impacts the accuracy of object edge segmentation in the images. To address this issue, we propose an improved FCUNet network architecture. FCUNet modifies the basic framework based on the UNet-V3 model. In the encoder, we replace the original pooling operation with a bilinear interpolation layer and introduce a fractal curve-based module to enhance both the accuracy and efficiency of feature extraction. In the decoder, we refine the multi-channel feature maps from different layers to optimize feature representation. These improvements are aimed at increasing the model’s adaptability to complex remote sensing scenarios, particularly when handling object edges and intricate details. The overall structure of the FCUNet is shown in Figure 1, designed specifically with the unique challenges of remote sensing image processing in mind.

### 2.1. Encoder with Combined Bilinear Interpolation Sampling and Fractal Curve Module

Most downsampling methods in the encoder–decoder framework of U-shaped models rely on pooling operations, with MaxPool and AveragePool being the most commonly used. However, MaxPool disregards the influence of the remaining N^2^-1 values, limiting the model’s full utilization of feature layer information, while AvgPool fails to differentiate the weights of the values. In contrast, bilinear interpolation is a method of interpolating an image on a two-dimensional plane. It estimates the value of a target point by weighted averaging the four nearest points, which form a rectangular region within which the target point is located. This approach fully considers both the values of the points in the region and the weights of each point. Therefore, in the downsampling stage of the FCUNet encoder, we introduce bilinear interpolation to enhance feature extraction accuracy while preserving spatial information. Additionally, this technique smooths the image, reducing the appearance of jagged edges, and adapts well to different feature scales. Figure 2 illustrates the operational processes and outcomes of these three downsampling methods, where “pixel position” denotes the position of the postsampling point before sampling.

For each pixel point after downsampling, we consider its four nearest neighbors surrounding its corresponding position in the original image. These four points form a rectangular region, with the target point located inside this rectangle. First, we apply linear interpolation along the horizontal direction, followed by linear interpolation in the vertical direction, yielding the final interpolated result. The calculation process is illustrated in Figure 3, which visualizes the positional relationship between the target point and the four surrounding points. The mathematical expression for this process is provided in Equation (1).
(1)$$\begin{array}{l} f(P)=(x_{2}-x)(y_{2}-y)f(Q_{11})+(x-x_{1})(y_{2}-y)f(Q_{21})\\ \;\;\;\;+(x_{2}-x)(y-y_{1})f(Q_{12})+(x-x_{1})(y-y_{1})f(Q_{22}) \end{array}$$

Formula explanation:
P(x,y): The estimated value of the target point.$Q_{11}\left(x_{1},y_{1}\right),Q_{12}\left(x_{1},y_{2}\right),Q_{21}\left(x_{2},y_{1}\right),andQ_{22}\left(x_{2},y_{2}\right)$: The values of the four nearest neighbor pixels.f(Point): The weight of the point.

In the encoder, we introduce a fractal curve-based module to traverse multi-dimensional feature maps at various scales. Fractal curves of different orders can efficiently traverse feature maps of different resolutions. The appropriate order of the fractal curve is selected based on the input image size, balancing computational complexity with traversal accuracy. This process can be understood as mapping a two-dimensional image into a one-dimensional space while preserving the spatial relationships between pixels. The resulting one-dimensional pixel sequence contains all the information of the original image, and the sequence is then reconstructed back into the original two-dimensional form—this process is referred to as “straightening and reconstruction”. By considering both the original image dimensions and the properties of the fractal curves, each element in the sequence can be returned to its original position via inverse mapping, resulting in interesting transformation effects. This study focuses on five common fractal curves and their application to pixel traversal: the Quadratic Gosper Curve (QGC), Square Sierpinski Curve [36] (SSC), Peano Curve [37] (PC), Krishna Anklets Curve (KAC), and Hilbert curve [38] (HC). Figure 4 provides a schematic of how these five curves traverse pixels in a digital image.

By examining the traversal paths of these five fractal curves, we observe that each curve has its own strengths and limitations: QGC and SSC suffer from pixel omission during traversal, leading to their exclusion from further consideration. PC can complete the traversal but requires that the image slices be in $3^{m}$ square formats, and its locally extended traversal paths limit its applicability. KAC encounters pixel repetition during traversal, which can result in errors in image size during the straightening and reconstruction process. The wander path diagrams in Figure 4 highlight these prominent drawbacks for each curve.

To fully evaluate the five fractal curves, we conducted a detailed analysis of their differences and summarized the results in Table 1. After a thorough review, we selected the Hilbert curve as the focus of this study due to its clear advantages in terms of completeness, efficiency, and applicability in pixel traversal. These qualities make it the most suitable choice for meeting our research requirements.

Fractal curve traversal paths can be generated using the L-system [39] method. In this thesis, we provide the pseudo-code for generating Hilbert curves based on the L-system, as detailed in Table 2. Additionally, we illustrate the recursive transformation of Hilbert curves from order 1 to 5, as drawn by the L-system. Figure 5 presents a series of images visualizing the structural characteristics and increasing complexity of Hilbert curves across different orders.

The pixel wandering paths generated by the Hilbert curve of order m are tailored for image slices with dimensions 2m×2m. For our experiments, we chose m=8. Algorithm 1 outlines the steps involved in this process. To demonstrate the impact of a fractal curve on an image slice, we present the effect of a Hilbert curve of order 4 applied to a 16 × 16 square slice. The pixel chain generated after traversal is then reconstructed back to the original image size, and the comparison before and after the Hilbert curve is applied to the digital template is shown in Figure 6. Additionally, the FC Encode and FC Decode modules in FCNet represent a pair of inverse operations, as illustrated in Figure 7.
**Algorithm 1** Multi-channel feature map walk traversal reconstruction of Hilbert curveInput: Feature map F∈RC×H×w, Order m∈RStep 1: The position coordinates vector for the walk-through path of an m-order Hilbert curve is generated as x∈R1×N and y∈R1×N.Step 2: Matrix the multi-channel feature map, and extract the information at the points according to the synchronized x and y coordinates.Step 3: Arrange the information points into a sequence to obtain Seqpixel∈RC×HW.Step 4: Reconstruct Seqpixel into F′∈RC×H×w.Output: F′


### 2.2. Decoder with Combined Aggregation Layers

The UNet series integrates multi-scale information but typically focuses on processing feature layers at the same scale, or does not differentiate well between layers, resulting in a lack of fine-grained processing. To address this, FCUNet enhances feature map processing by applying L1 regularization to remove non-same-scale feature layers, and then concatenating them with the same-scale layers. FCUNet introduces a feature selection and fusion strategy, referred to as the Agg layer, with the detailed implementation shown in Figure 8.

Firstly, the multi-dimensional feature map outputs from other layers that are not in the same layer are aggregated, and the dimension N of the same layer feature values is obtained. Then, the L1 normal form value of a single feature map is calculated, and multiple normal form values are sorted from large to small. The top N values corresponding to the feature maps are selected and combined with the multi-dimensional feature maps in the same layer to form a 2N dimensional feature map as input for other modules. The advantage of this method is that it maximizes the utilization of feature information by selectively fusing layers. By using L1 regularization, it ensures the selection of representative feature layers, reduces computational complexity, and minimizes the number of training parameters in the decoder, contributing to a more lightweight model.

By incorporating the Agg layer, FCUNet effectively integrates features across different layers while preserving crucial semantic information. This approach is particularly well suited for handling complex scenes and fine feature boundaries in remote sensing images. Compared to UNet and UNet3+, FCUNet’s Agg layer offers a more refined and efficient feature fusion mechanism.

### 2.3. Evaluation Metrics

The evaluation of semantic segmentation models relies on the computation of a confusion matrix. In binary classification, the confusion matrix includes four parameters: True Positive (TP), false negative (FN), False Positive (FP), and True Negative (TN), as outlined in Table 3. Since semantic segmentation operates at the pixel level, these parameters are computed based on the intersection and union of the target and prediction results. The specific region division and calculation methods are illustrated in Figure 9.

Classical semantic segmentation models often improve evaluation metrics by increasing the TP value. However, this approach can also lead to a rise in FP, which limits the overall improvement of the final evaluation metrics. To address this issue, our study introduces a module based on fractal curves, designed to increase TP while effectively reducing the FP region and its corresponding values. The strength of this approach lies in maintaining a high recognition rate for correctly segmented regions (TP) while minimizing the regions falsely identified as positive examples (FP) through a fine pixel traversal strategy. This ensures the accuracy and reliability of the segmentation results while enhancing the overall model performance. The method achieves a more balanced and superior performance in semantic segmentation tasks by not only improving the correct recognition rate but also effectively controlling misclassifications.

In this experiment, we utilized multiple metrics to comprehensively evaluate the segmentation performance of the model. These metrics include Intersection over Union (IoU), Mean Intersection over Union (MIoU), Mean Pixel Accuracy (MPA), and the F1-score, each reflecting a different aspect of the model’s performance. The formulas for these metrics are as follows:*IoU* (Intersection over Union): IoU represents the ratio of the intersection to the union of the ground truth and predicted segments. The formula for IoU is presented in Equation (2).
(2)$$IoU=\frac{Target\cap Prediction}{Target\cup Prediction}=\frac{TP}{FP+TP+FN}$$

*MIoU* (Mean Intersection over Union): MIoU is an extension of the IoU metric used for multi-class image segmentation tasks. The formula for MIoU is presented in Equation (3).


(3)
$$MIoU=\frac{1}{k+1}\sum_{i=0}^{k}\frac{TP}{TP+FN+FP}$$


*MPA* (Mean Pixel Accuracy): MPA measures the average pixel accuracy across all classes. The formula for MPA is presented in Equation (4).


(4)
$$MPA=\frac{1}{k+1}\sum_{i=0}^{k}\frac{TP}{TP+FP}$$


*F1-Score*: The F1-score is the harmonic mean of precision and recall. The formula for F1 is presented in Equation (5).


(5)
$$F1=\frac{2\times Precision\times Recall}{Precision+Recall}=\frac{2\times \mathrm{TP}}{FP+2\times TP+FN}$$


Precision is calculated as Precision = TP/(TP + FP), and recall is calculated as Recall = TP/(TP + FN). The F1-score, which is the harmonic mean of precision and recall, provides a comprehensive measure of model performance.

Together, these metrics offer a thorough assessment of the model’s segmentation effectiveness. IoU and MIoU focus on the accuracy and completeness of the segmentation, MPA reflects pixel-level classification accuracy, and the F1-score balances precision and recall, providing an overall evaluation of the model’s performance. Analyzing these metrics collectively allows for a comprehensive evaluation of the model in various aspects, laying a solid foundation for further improvement and optimization.

Additionally, to specifically address edge segmentation quality, we designed an evaluation function called the adhesion coefficient $\varphi_{ac}$. This coefficient evaluates the performance changes before and after the module is inserted into the model, focusing on the clarity of edge segmentation between the predicted result (Prediction) and the ground truth (Ground Truth). The adhesion coefficient reflects the class-averaged number of misclassified pixels, where a lower value indicates better segmentation and more clearly defined object edges. The formula for the adhesion coefficient is shown in Equation (6).
(6)$$\varphi_{adhesion\_cofficient}=\frac{1-\frac{Pixel_{TP}}{Pixel_{img}}}{Classnum}$$
where $Pixel_{TP}$ denotes the sum of all the classes of the image that are categorized correctly, $Pixel_{img}$ represents the total number of pixels in the image, and $Class_{num}$ indicates the number of categories in the dataset.

In certain application scenarios, we may focus solely on the foreground objects in an image, excluding the background. For this situation, we provide a modified version of the equation, as shown in Equation (7).
(7)$$\varphi_{adhesion\_cofficient\_foregraound}=\frac{1-\frac{Pixel_{TP}}{Pixel_{img}-Pixel_{background}}}{Classnum}$$
where $Pixel_{background}$ denotes the number of pixels in the background. This modified version places greater emphasis on the segmentation quality of foreground objects.

The introduction of the adhesion coefficient allows us to more precisely quantify improvements in edge segmentation, providing targeted guidance for model optimization. It also facilitates fairer and more meaningful comparisons between different models and methods.

## 3. Results and Discussion

### 3.1. Experimental Datasets and Platform Parameter Settings

The WHDLD dataset is a high-resolution remote sensing image collection acquired by Wuhan University using the Gaofen-1 and ZiYuan-3 satellites. It consists of 4940 satellite images of Wuhan city, accompanied by their corresponding semantic segmentation masks. Each image has a spatial resolution of 2 m and measures 256×256 pixels. Both the raw and labeled images are in a three-channel format with an 8-bit depth.

The dataset categorizes features into six classes: buildings, roads, sidewalks, vegetation, bare soil, and water bodies. This classification encompasses the major feature types found in urban environments, providing a comprehensive sample for urban remote sensing studies. Since the dimensions of the original and masked images meet the requirements of our study, they can be used directly without additional preprocessing steps. Figure 10 displays examples of the images and masks from the dataset, along with the corresponding RGB color table for each category. This visualization not only intuitively illustrates the dataset’s content but also serves as a clear reference for subsequent analysis and processing.

The Potsdam and Vaihingen datasets are high-resolution remote sensing image collections obtained through satellite photography. These datasets cover the German cities of Potsdam and Vaihingen, respectively, and feature an exceptionally high spatial resolution of 5 cm. In total, the datasets contain 38 images with a resolution of 6000 × 6000 pixels; the Potsdam dataset consists of images with a fixed size, while the Vaihingen dataset includes images of varying dimensions. Both the raw and labeled images are in a three-channel format with an 8-bit depth.

To facilitate processing and maintain consistency, we uniformly resized all the images to 256×256 pixels. After processing, the two datasets consist of 20,000 images each, from which 10,000 images are extracted to form the training and testing sets in a 7:3 ratio. This preprocessing not only standardizes the data but also enhances their suitability for training and testing deep learning models. Figure 11 displays the sample images from both datasets, allowing us to visualize their characteristics, including detail performance at high resolution and the feature characteristics of different urban environments. The experimental environment for the model is detailed in Table 4.

### 3.2. Experimental Result Analysis

In the FCUNet network framework illustrated in Figure 5, the inputs and outputs of FCEncode and FCDecode are represented by dashed lines, indicating that the insertion points of these two modules within FCUNet are variable. The choice of insertion points can significantly affect the final evaluation metrics, necessitating experiments for each potential insertion position. Given that the encoder component of the U-shaped framework typically consists of four layers, there are four possible insertion points.

To identify the optimal insertion points for FCEncode and FCDecode, we conducted experiments using the UNet3+ model and the WHDLD dataset. The experimental results are presented in Figure 12: column a displays the original 256×256 resolution remote sensing images, column b shows the corresponding ground truth images, and columns c-f illustrate the segmentation effects when FCEncode and FCDecode are inserted into the 1st to 4th layers of UNet3+, respectively.

From the results, it is evident that as the insertion layer increases, the yellow area in the first row gradually aligns more closely with the ground truth. However, in the second row, none of the segmentation results successfully identify the playground as depicted in the ground truth. In the third row, as the insertion layer increases, the boundaries of the gray area within the box become clearer. Notably, in column e (third layer insertion), the gray area appears as a large patch, obscuring the middle sidewalk entirely.

These observations underscore the significant impact of the insertion positions of FCEncode and FCDecode on the segmentation outcomes, providing a solid foundation for selecting the optimal insertion point.

Table 5 presents the MIoU, MPA, F1-score, and φ values for four different insertion positions within UNet3+. The data reveals that as the insertion layer for FCEncode and FCDecode increases, all four evaluation metrics show an upward trend, peaking at the fourth layer. Specifically, when inserted at this layer, MIoU reaches 52.48%, MPA reaches 71.12%, F1 reaches 65.92%, and the φ value is 3.72. These results clearly indicate that the model’s performance metrics are optimal when the FCEncode and FCDecode modules are positioned at the fourth layer. Based on this finding, we will maintain the insertion of these two modules at the fourth layer in the subsequent experiments to ensure the model achieves its best performance.

Next, we conducted ablation experiments to verify that integrating the fractal curve-based module significantly enhances model performance compared to its absence. Following this, we selected original semantic segmentation models, including UNet3+, U^2^Net, UNeXt, and GCtx-Unet, and performed comparative experiments on the WHDLD dataset. The experimental results are displayed in Figure 13, which consists of multiple columns to illustrate the segmentation effects of the different models. Column (a) shows the original 256×256 resolution remote sensing images, while column (b) presents the corresponding ground truth images. Columns (c) and (d) display the segmentation results of UNet3+ without and with the fractal curve module, respectively. Similarly, columns (e) and (f) show the segmentation results of U^2^Net in both scenarios. Column (g) presents the segmentation results of the UNeXt network, column (h) shows the results from the GCtx-UNet network, and column (i) depicts the segmentation results of our proposed model. Through these comparisons, we can visually observe the performance differences among the various models in handling complex remote sensing images, particularly highlighting the performance improvements provided by the fractal curve module in the existing models while also showcasing the advantages of our proposed model.

From the segmentation results in Figure 13, we can observe performance differences across the various models. The segmentation results from the UNet3+ network (column c) indicate that roads and pavements, as well as roads and bare soil, are frequently misclassified, leading to significant error areas where bare soil and roads are misidentified as shrubs (highlighted in green). This issue primarily arises from the sparse sampling points in the feature layers during the downsampling stage in the UNet3+ model, which weakens the model’s representation capabilities. In contrast, the model integrated with the fractal curve module (column d) demonstrates a substantial improvement in the classification of larger areas. This enhancement is attributed to the rearrangement of pixel points in the multi-channel feature map in the deeper layers, causing similar pixels to cluster more effectively.

A similar improvement is evident in the U^2^Net model, where the results with the fractal curve module (column f) outperform those without it (column e), thereby reducing the likelihood of misclassifying bare soil as shrubs. By comparing columns (c), (e), (g), (h), and (i) in Figure 13, it is clear that the segmentation performance of the FCUNet model surpasses that of the other four models. The FCUNet network utilizes FCEncode and FCDecode to enhance the clustering of pixel points in the multi-channel feature map, thereby increasing the aggregation of similar pixels. Furthermore, the aggregation layer effectively utilizes features at different scales and incorporates fractal features, which improves segmentation accuracy for multi-class remote sensing images. Compared to the other models, FCUNet achieves clearer and smoother segmentation edges when dealing with large-scale objects, significantly reducing the mis-segmentation caused by complex backgrounds, shadows, and closely spaced objects that lead to boundary adhesion.

Overall, the segmentation results of FCUNet outperform those of the UNet3+, U^2^Net, UNeXt, and GCtx-Unet models. These qualitative observations are further supported by the quantitative performance metrics detailed in Table 6.

In the table, the first four models are compared through ablation experiments, demonstrating that the models incorporating the fractal curve module outperform those without it. The highest values for MIoU, MPA, F1-score, and the adhesion coefficient φ are achieved by the models with the fractal curve module, reaching 52.48%, 71.12%, 65.92%, and 3.72%, respectively, effectively validating the module’s effectiveness. Additionally, the experimental results for the FCUNet model exceed those of the other networks. The aggregation layer contributes to a reduction in the decoder’s parameters, resulting in a lightweight model. For FCUNet, the four evaluation metrics reach 54.71%, 74.43%, 67.57%, and 3.45%, confirming its superior performance. Figure 14 presents the confusion matrix comparing FCUNet’s inferred masks with the ground truth, where it is evident that the pixel segmentation accuracy for vegetation and water is relatively high. However, the segmentation performance for smaller areas, such as roads and bare soil, is comparatively lower.

In this paper, experiments were conducted on both the Potsdam and Vaihingen datasets, with the segmentation results presented. While these datasets, like WHDLD, fall under the category of remote sensing images, they exhibit distinct characteristics. Compared to WHDLD, where objects are often small and have complex overlaps, the objects in Potsdam and Vaihingen typically cover larger areas, with fewer overlaps and lower complexity.

The segmentation results for Potsdam are displayed in Figure 15. In columns (d) and (f), where yellow represents cars, the segmentation boundaries are notably clearer than in columns (c) and (e). Additionally, the models incorporating the fractal curve module demonstrate better segmentation of the large red background areas compared to those without it. When comparing FCUNet with the four original models, FCUNet exhibits clearer boundaries for cars and fewer misclassifications of the other background categories.

Figure 16 presents the segmentation results for Vaihingen. In the second row of columns (c) and (e), the green areas representing trees exhibit boundary adhesion, which is reduced in columns (d) and (f) with the fractal curve module. Moreover, in the third row, the model with the fractal curve module lowers the probability of misclassifying low vegetation as impervious surfaces. Similarly, FCUNet’s segmentation results surpass those of the other four original models, showing fewer false negatives for buildings (represented in blue) and a lower false detection rate for low vegetation.

The quantitative performance indicators for the Potsdam and Vaihingen datasets are summarized in Table 7 and Table 8, respectively. Consistent with the quantitative analysis for WHDLD, the models with the fractal curve module outperform those without in the Potsdam experiments, with FCUNet excelling in all four evaluation metrics compared to the original models. On the Potsdam dataset, MIoU, MPA, and F1-score reach 63.15%, 75.79%, and 78.55%, respectively, while $\varphi_{ac}$ is 2.61%. For the Vaihingen dataset, these metrics are 50.20%, 63.25%, and 59.90%, and $\varphi_{ac}$ is 1.94%.

The confusion matrix for Potsdam (Figure 17) reveals that pixel classification for impervious surfaces, low vegetation, and clutter is relatively strong, while buildings and trees demonstrate poorer classification performance. In the confusion matrix for Vaihingen (Figure 18), pixel classification for impervious surfaces, buildings, low vegetation, and trees is good, whereas cars and clutter exhibit weaker classification performance.

## 4. Conclusions

The acquisition of remote sensing images is influenced by various factors, often leading to challenges such as blurred boundary segmentation in the existing deep semantic segmentation networks. To address this issue, this study proposes the FCUNet multi-class segmentation model, which incorporates a fractal curve module. Building on the encoder structure, the model employs bilinear interpolation instead of the traditional pooling operations and introduces the fractal curve module to traverse the multi-channel feature map, aiming to aggregate individual object pixels effectively. This design seeks to enhance pixel aggregation, thereby improving the segmentation of both large and small objects in complex environments. In the decoder section, the model utilizes an aggregation layer to weigh and filter the different multi-channel feature layers. The experiments are divided into two parts: first, the ablation experiments demonstrate that the model with the fractal curve module shows improvements across all the evaluation metrics, confirming the module’s effectiveness. Second, the comparative analysis of FCUNet against the four original semantic segmentation models reveals superior performance in multi-class segmentation tasks across three remote sensing image datasets, further validating the model’s capabilities in semantic segmentation.

The high clustering and continuity of the Hilbert curve make it an ideal tool for optimizing the encoding and indexing of spatial data, thereby improving the rationality of urban planning and the recovery and loss statistics of buildings. At the same time, in the storage and indexing of spatiotemporal big data, the combination of the Hilbert curve with database technology is expected to solve the storage and query efficiency problems of large-scale data, which is of great significance in addressing big data challenges; the three-dimensional curve helps to solve complex spatial filling problems, improving the efficiency and accuracy of multi-dimensional data encoding.

## Figures and Tables

**Figure 1 sensors-24-07159-f001:**
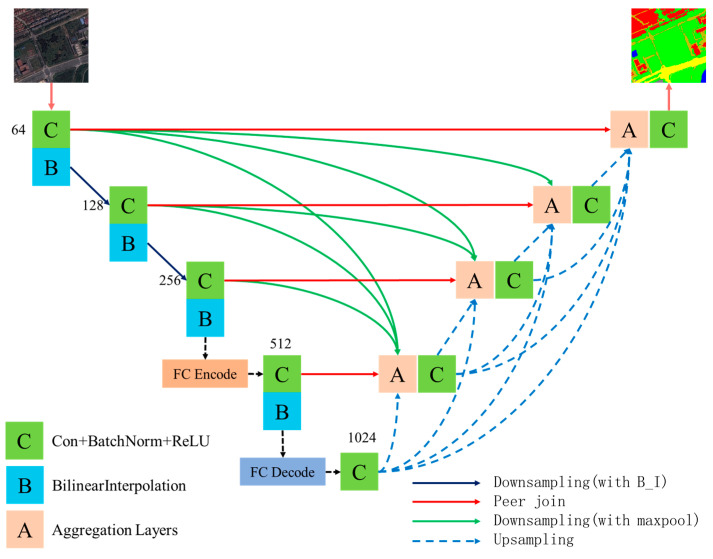
FCUNet model framework.

**Figure 2 sensors-24-07159-f002:**
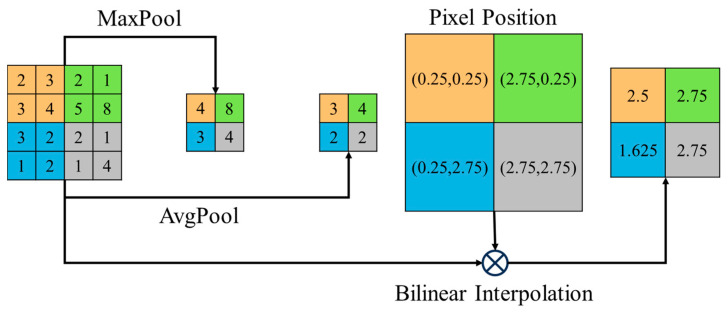
Pooling and bilinear interpolation calculation process.

**Figure 3 sensors-24-07159-f003:**
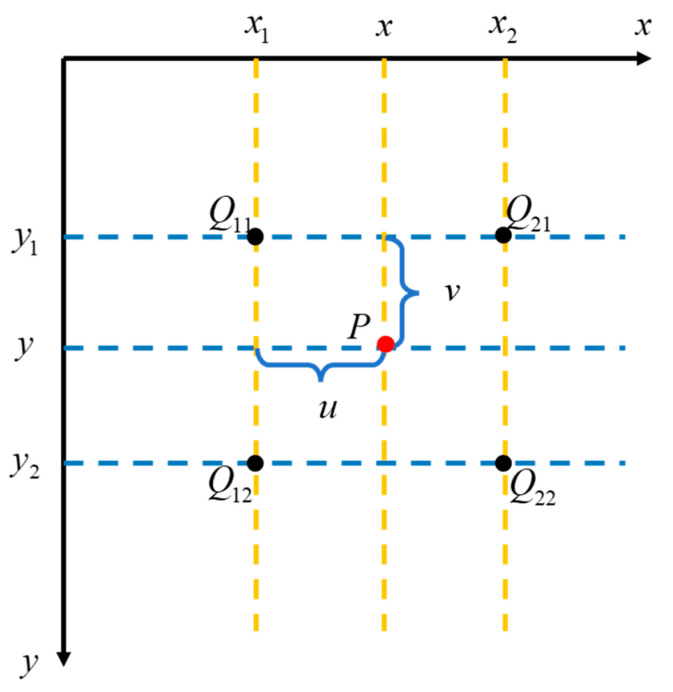
Bilinear interpolation calculation diagram.

**Figure 4 sensors-24-07159-f004:**
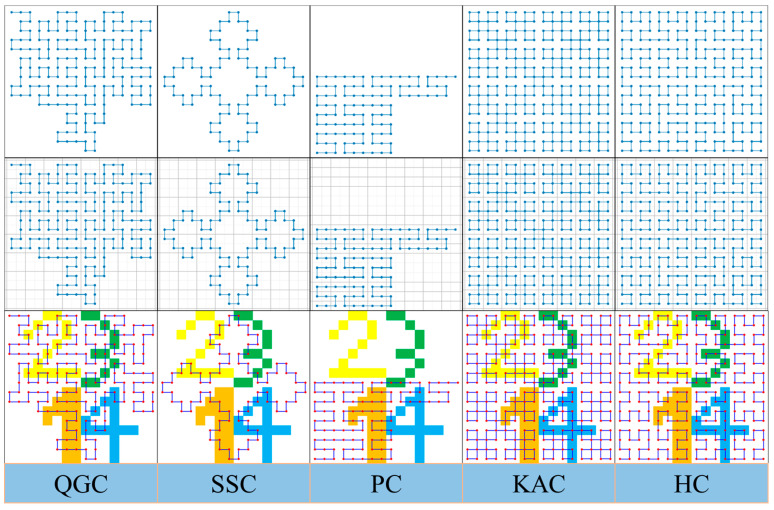
Traversal path of 4th-order (**top**) grid-based traversal path (**middle**) example (**bottom**).

**Figure 5 sensors-24-07159-f005:**

Generation of Hilbert curves at different orders (1–5).

**Figure 6 sensors-24-07159-f006:**
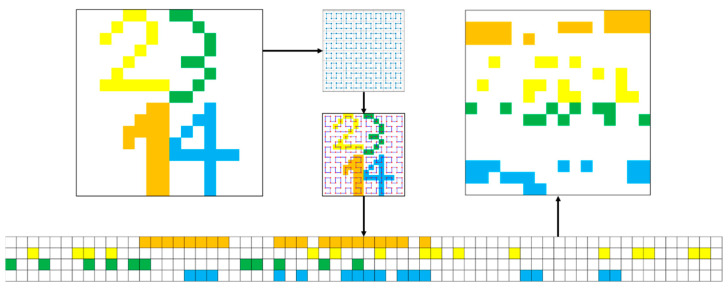
Before and after applying the Hilbert curve to the digital template.

**Figure 7 sensors-24-07159-f007:**
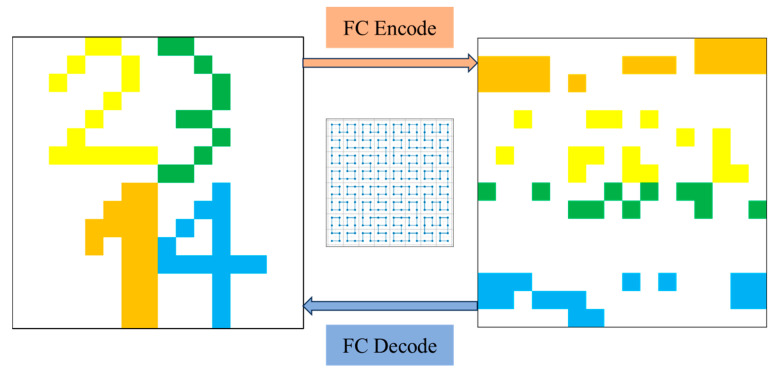
FC Encode and FC Decode modules.

**Figure 8 sensors-24-07159-f008:**
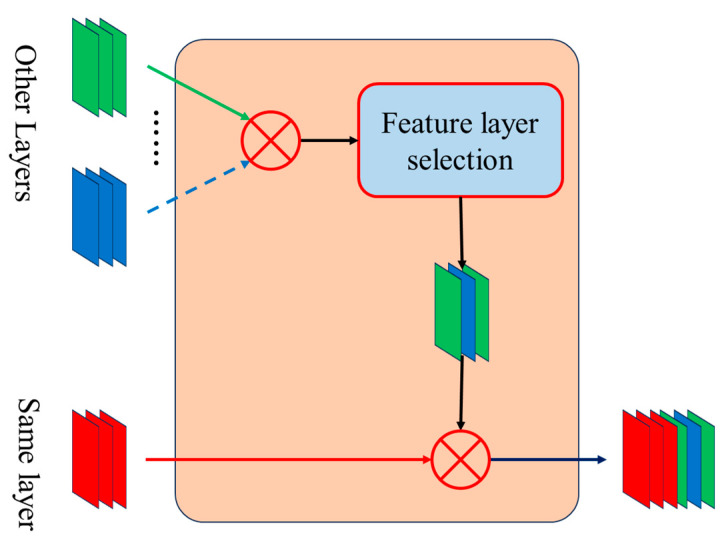
Aggregation layer processing diagram.

**Figure 9 sensors-24-07159-f009:**
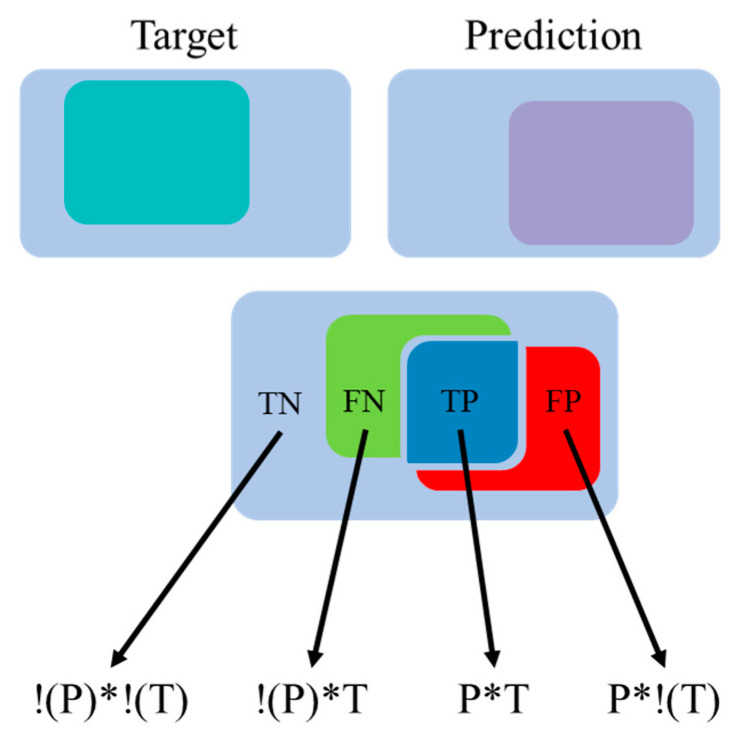
Corresponding areas and calculation formulas for target and prediction.

**Figure 10 sensors-24-07159-f010:**
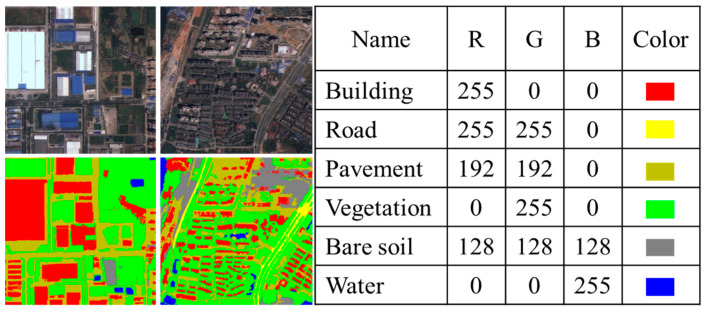
Examples of the WHDLD dataset and RGB correspondence table.

**Figure 11 sensors-24-07159-f011:**
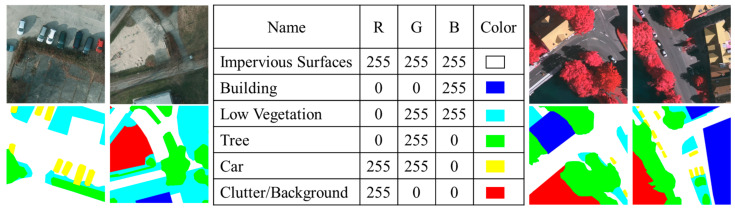
Examples and RGB correspondence table for the Potsdam (**left**) and Vaihingen (**right**) datasets.

**Figure 12 sensors-24-07159-f012:**
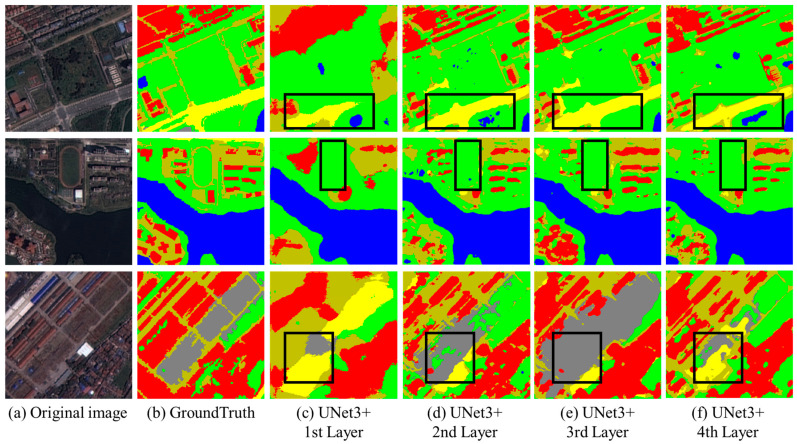
Segmentation effects of the fractal curve module at different insertion positions.

**Figure 13 sensors-24-07159-f013:**
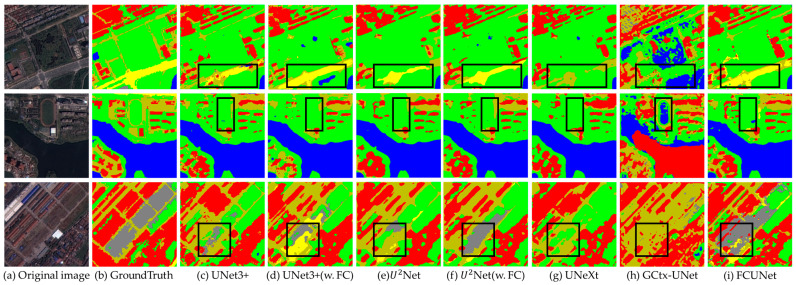
Comparison of WHDLD segmentation results.

**Figure 14 sensors-24-07159-f014:**
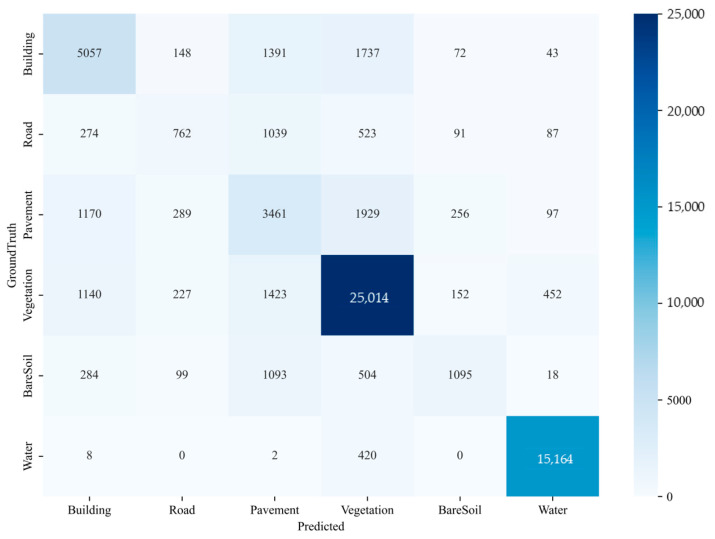
Confusion matrix of FCUNet model on the WHDLD dataset.

**Figure 15 sensors-24-07159-f015:**
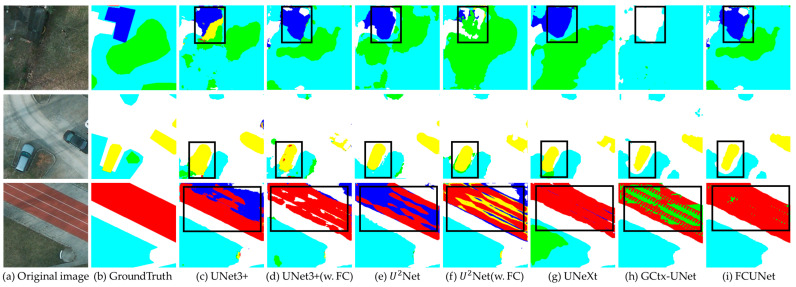
Comparison of Potsdam segmentation results.

**Figure 16 sensors-24-07159-f016:**
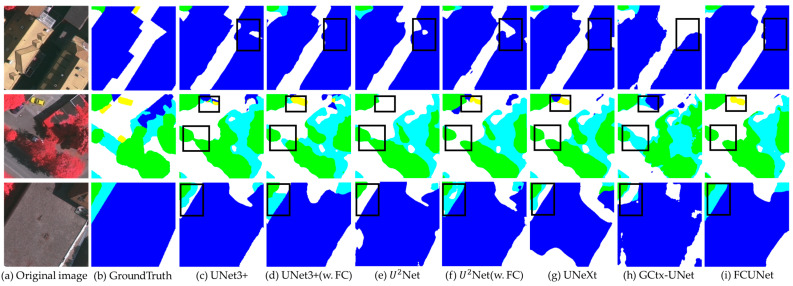
Comparison of Vaihingen segmentation results.

**Figure 17 sensors-24-07159-f017:**
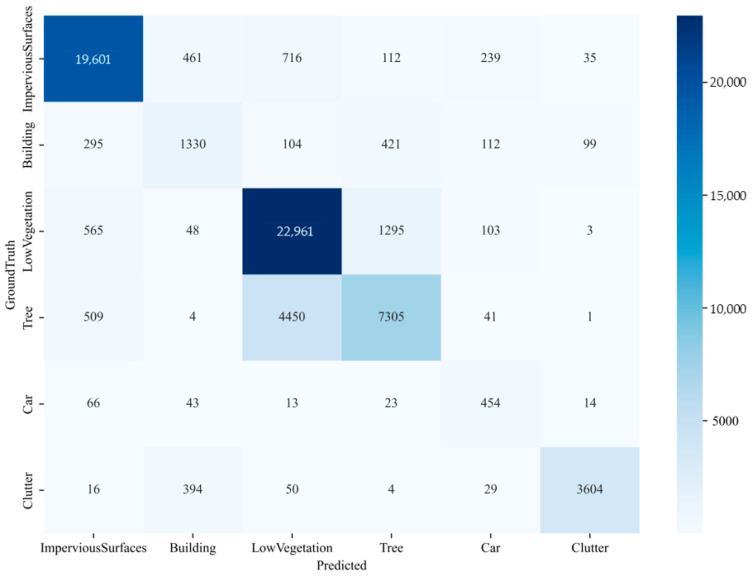
Confusion matrix of FCUNet model on the Potsdam dataset.

**Figure 18 sensors-24-07159-f018:**
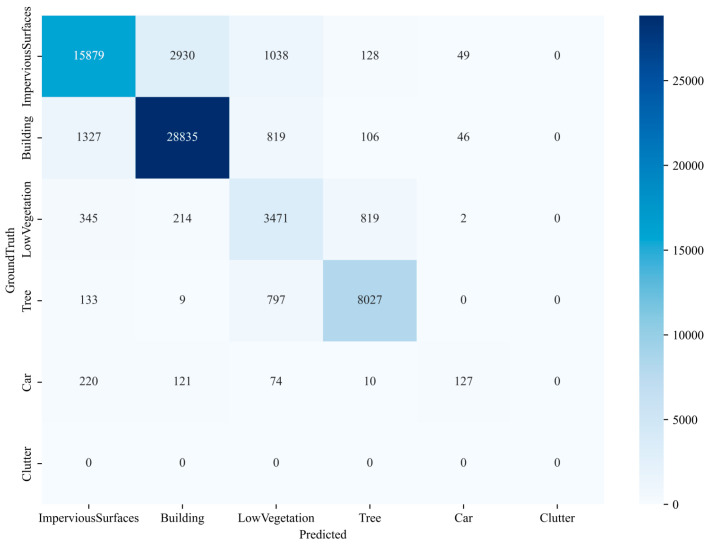
Confusion matrix of FCUNet model on the Vaihingen dataset.

**Table 1 sensors-24-07159-t001:** Differences among the five fractal curves.

Curve	Advantages	Disadvantages
QGC	Simple rules, good self-similarity, and high filling efficiency	Can only represent one-dimensional fractal structures with limited variation; exhibits missing pixel points
SSC	Strong self-similarity, infinitely divisible, and easy to construct	Uneven gap distribution, lower filling efficiency, and exhibits missing pixel points
PC	Can fill a two-dimensional plane, good self-similarity, and infinitely divisible Image slices must be squares with dimensions of a power of 3	Complex construction rules and many turns and changes, with repeated pixel traversal
KAC	Simple construction and good self-similarity	One-dimensional curve and high computational complexity and cost under high detail Repeated pixel traversal Traversal path is recursively too long
HC	Self-similar recursion, compact and gapless filling, and mapping preserves locality	Many turns, complex construction, and high computational cost for higher-order details

**Table 2 sensors-24-07159-t002:** L-System generation expression for the Hilbert curve.

Representation as Lindenmayer System
Alphabet:A,B Constants:F + - Axiom:A Production rules: A→+BF-AFA-FB+ B→-AF+BFB+FA-
F:draw forward +:turn left 90° -:turn right 90° A/B:ignored during drawing

**Table 3 sensors-24-07159-t003:** Explanation of parameters in a binary classification confusion matrix.

Confusion Matrix		Predicted	
		Positive	Negative
GroundTruth	Positive	TP (True Positive)	FN (False Negative)
	Negative	FP (False Positive)	TN (True Negative)

**Table 4 sensors-24-07159-t004:** Experimental environment.

Platform	Name
CPU	Intel(R)Xeon(R)Gold6240/32G
GPU	Tesla T4/32G
Disk capacity	HDD/900G
The operating system	Ubuntu18
Deep learning framework	Pytorch1.13

**Table 5 sensors-24-07159-t005:** Evaluation metrics for the fractal curve module at different insertion positions.

	MIoU	MPA	F1-Score	φ
1st Layer	37.39	57.29	50.28	5.78
2nd Layer	49.98	68.77	63.46	3.85
3rd Layer	50.58	63.88	63.95	3.96
**4th Layer**	**52.48**	**71.12**	**65.92**	**3.72**

**Table 6 sensors-24-07159-t006:** Model experiment data on the WHDLD dataset.

Model	MIoU	MPA	F1-Score	$\varphi_{ac}$
UNet3+	45.29	66.54	56.75	3.86
UNet3+ (w. FC)	52.48	71.12	65.92	3.72
U^2^Net	43.07	71.95	53.91	4.00
U^2^Net (w. FC)	44.31	71.86	55.79	3.89
UNeXt	37.10	43.57	44.79	4.41
GCtx-Unet	19.38	31.07	31.20	10.14
**FCUNet**	**54.71**	**74.43**	**67.57**	**3.45**

**Table 7 sensors-24-07159-t007:** Model experiment data on the Potsdam dataset.

Model	MIoU	MPA	F1-Score	$\varphi_{ac}$
Unet3+	50.58	60.27	61.11	3.48
UNet3+ (w. FC)	54.93	72.20	68.23	3.22
U^2^Net	50.34	76.67	62.59	3.27
U^2^Net (w. FC)	50.67	68.77	64.75	3.03
UNeXt	22.67	27.22	29.62	6.55
GCtx-Unet	22.10	35.71	29.17	6.67
**FCUNet**	**63.15**	**75.79**	**75.55**	**2.61**

**Table 8 sensors-24-07159-t008:** Model experiment data on the Vaihingen dataset.

Model	MIoU	MPA	F1-Score	$\varphi_{ac}$
Unet3+	45.52	58.58	53.43	2.46
UNet3+(w. FC)	47.32	62.89	56.79	2.68
U^2^Net	47.96	54.31	55.16	2.20
U^2^Net(w. FC)	48.54	72.67	55.75	2.06
UNeXt	45.74	53.39	53.63	2.57
GCtx-Unet	37.69	45.71	46.68	4.13
**FCUNet**	**50.20**	**63.25**	**59.90**	**1.94**

## Data Availability

The experimental data presented in this paper are available upon request from the corresponding author.

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
