# Peer review of "A Fractal Curve-Inspired Framework for Enhanced Semantic Segmentation of Remote Sensing Images"

_sensors, 2024, doi:10.3390/s24227159_

Round 1
Reviewer 1 Report
Comments and Suggestions for Authors
1. The relationship between the current situation analysis and the problem to be solved in the Introduction is not close enough, failing to fully introduce the problem to be solved. Additionally, issues such as scene clutter, lighting conditions, or cloud cover affecting edge features are analyzed, but these are not related to the subsequent dataset,then suggest introducing their correlation. The English could be improved to more clearly express the research.
2. The Methods section does not describe which basic network architecture the research is based on and improved upon. Are there comparative experiments with the basic network structure?
3. In Figure 2, the bilinear interpolation calculation process and formula are not clearly explained. The meanings of different variables in the formula are ambiguous, such as the x-axis and x variable. All variable definitions in the paper should ensure uniqueness.
4. Combine Figure 4 and Table 1 to describe the shortcomings of various fractal curves and provide reasons for choosing the Hilbert curve. What are the differences before and after applying HC in Figure 6? What problems can it solve, and how does it relate to the problem addressed in this paper?
5. The role of the FC encoding and decoding process in the overall network structure in Figure 7 is unclear. In Figure 1, the FC encoding and decoding process is not a simple loop, but in Figure 7, the meaning of the simple loop is unclear.
6.The introduction of some basic knowledge of evaluation metrics is a bit redundant and should be streamlined.
7. The first paragraph after Section 3.1 is unrelated to the main text. In the third section, the introduction of the dataset should explain the composition of the training and test sets. The Potsdam and Vaihingen datasets only include 38 images. Details on how to augment them for training and testing the network model should be provided to avoid overfitting due to the small data size.
8. Introduce future research directions based on the work of this paper.
9.Check the format of the 16th literature and examine similar issues.
Author Response
Comments 1: The relationship between the current situation analysis and the problem to be solved in the Introduction is not close enough, failing to fully introduce the problem to be solved. Additionally, issues such as scene clutter, lighting conditions, or cloud cover affecting edge features are analyzed, but these are not related to the subsequent dataset, then suggest introducing their correlation. The English could be improved to more clearly express the research.
Response 1:Thank you for your careful review and thoughtful suggestions. The introduction section has been revised by removing irrelevant statements and adding the connection between medicine and remote sensing for semantic segmentation, as shown in L86-L99.
Comments 2: The Methods section does not describe which basic network architecture the research is based on and improved upon. Are there comparative experiments with the basic network structure?
Response 2:Thanks for your instruction. FCUNet has been modified based on the UNet-V3 model framework, and comparative experiments can be viewed in Result and Discussion.
Comments 3: In Figure 2, the bilinear interpolation calculation process and formula are not clearly explained. The meanings of different variables in the formula are ambiguous, such as the x-axis and x variable. All variable definitions in the paper should ensure uniqueness.
Response 3:Thanks for your instruction. The principle and formula parameter explanation of bilinear interpolation have been added, as shown in L138-L141 and L161-L165.
Comments 4: Combine Figure 4 and Table 1 to describe the shortcomings of various fractal curves and provide reasons for choosing the Hilbert curve. What are the differences before and after applying HC in Figure 6? What problems can it solve, and how does it relate to the problem addressed in this paper?
Response 4:Thanks for your instruction. HC stands for Hilbert Curve. Figure 4 and Table 1 are combined to explain the advantages and disadvantages of different curves. The application of HC in Figure 6 is to adaptively adjust the order of the curve based on the size of different layer feature maps. Solving the problem of mutual adhesion of segmented objects in semantic segmentation.
Comments 5: The role of the FC encoding and decoding process in the overall network structure in Figure 7 is unclear. In Figure 1, the FC encoding and decoding process is not a simple loop, but in Figure 7, the meaning of the simple loop is unclear.
Response 5:Thanks for your instruction. FC Encode and FC Decoder in Figure 1 are two modules in the model. FC Decoder is the inverse operation of FC Encode, responsible for decoding the curve encoded image. The FC Encode in Figure 7 performs the process shown in Figure 6.
Comments 6: The introduction of some basic knowledge of evaluation metrics is a bit redundant and should be streamlined.
Response 6:Thanks for your instruction. The relevant description sentences for the evaluation indicators have been simplified, as shown in L271-L280.
Comments 7: The first paragraph after Section 3.1 is unrelated to the main text. In the third section, the introduction of the dataset should explain the composition of the training and test sets. The Potsdam and Vaihingen datasets only include 38 images. Details on how to augment them for training and testing the network model should be provided to avoid overfitting due to the small data size.
Response 7:Thanks for your instruction. Indeed, irrelevant paragraphs have been deleted; The Potsdam and Vaihingen datasets consist of 38 images with 6000x6000 pixels, which, when cut into a size of 256x256, do not overfit due to their small size, as shown in L326 and L331-L332.
Comments 8: Introduce future research directions based on the work of this paper.
Response 8:Thanks for your instruction. The future research directions on Hilbert curves have been added, as shown in L496-L503.
Comments 9: Check the format of the 16th literature and examine similar issues.
Response 9:Thanks for your instruction. The citation format has been modified.

Reviewer 2 Report
Comments and Suggestions for Authors
1. Bilinear interpolation and confusion matrix are very common, there is no need to introduce it too much.
2. The method proposed in this paper is not only applicable to remote sensing images, but can also be experimentally validated in other scenarios.
3. Max pooling or average pooling are common ways, can bilinear interpolation really improve performance? which has not been verified in the paper, and it is suggested to be analyzed during ablation experiments.
4. What's the benefits of the fractal curve? since it just rearrange all pixels, and the structural information of the image has been disrupted. Potential reasons should be analyzed.
5. How is the non-same scale feature layer removed by applying L1 regularization in the proposed Aggregation layer? This part needs to be described in more detail.
Author Response
Comments 1: Bilinear interpolation and confusion matrix are very common, there is no need to introduce it too much.
Response 1:Thank you for your careful review and thoughtful suggestions. We have done relevant simplification work on bilinear interpolation and confusion matrix, while retaining some core parts to cater to different readers.
Comments 2: The method proposed in this paper is not only applicable to remote sensing images, but can also be experimentally validated in other scenarios.
Response 2:Thanks for your instruction. Indeed, the inspiration for using curves mainly comes from one theory, which guarantees the information invariance of the transformation from one-dimensional curves to two-dimensional planes. As this paper introduces image processing, I am also researching its application in other scenarios.
Comments 3: Max pooling or average pooling are common ways, can bilinear interpolation really improve performance? which has not been verified in the paper, and it is suggested to be analyzed during ablation experiments.
Response 3:Thanks for your instruction. Indeed, pooling is a classic method for rapidly reducing the size of feature maps, but in my paper, I hope to use bilinear interpolation to make the numerical changes of feature maps less drastic and smoother.
Comments 4: What's the benefits of the fractal curve? since it just rearrange all pixels, and the structural information of the image has been disrupted. Potential reasons should be analyzed.
Response 4:Thanks for your instruction. The fractal curve is applied to the interlayer of the network. According to its curve theory analysis, it will transform the two-dimensional image into a one-dimensional pixel chain and then into a two-dimensional image, which is conducive to gathering information points in the feature map more, achieving the effect of gathering similar types and separating different types.
Comments 5: How is the non-same scale feature layer removed by applying L1 regularization in the proposed Aggregation layer? This part needs to be described in more detail.
Response 5:Thanks for your instruction. The relevant L1 paradigm deletion feature maps have been added to the paper, as shown in L228-L233.
